# Comment on Iida et al. Development of a New Method for Calculating Intraocular Lens Power after Myopic Laser In Situ Keratomileusis by Combining the Anterior–Posterior Ratio of the Corneal Radius of the Curvature with the Double-K Method. *J. Clin. Med.* 2022, *11*, 522

**DOI:** 10.3390/jcm11071996

**Published:** 2022-04-02

**Authors:** Ferdinando Cione, Maddalena De Bernardo, Nicola Rosa

**Affiliations:** Department of Medicine, Surgery and Dentistry, Scuola Medica Salernitana, University of Salerno, 84122 Salerno, Italy; nandocione1993@gmail.com (F.C.); nrosa@unisa.it (N.R.)

We read with great interest the article by Iida et al. [1] concerning an Intraocular Lens (IOL) power calculation method that combines the Anterior–Posterior (AP) ratio of the corneal radius of the curvature after Laser-Assisted in Situ Keratomileusis (LASIK). We would like to congratulate the authors on their paper because this is an important topic [2]. However, we would like to make some comments on points that, in our opinion, should be clarified.

(1)The Iida–Shimizu–Shoji (ISS) method is an interesting approach to solve the problem of IOL power calculation, but it was determined by the use of different instruments: in fact, the AP ratio of the corneal curvature was measured with the Pentacam Scheimpflug system, but Keratometry (K) values after surgery (Kpost) were obtained by the IOLMaster. This could generate a systematic error in the formula; actually, it has been demonstrated that when analyzing eyes with previous refractive surgery, data obtained from different machines cannot be used interchangeably [3].(2)We appreciate the authors cited the R Factor method [4]: this formula was the first that did not require preoperative parameters in IOL power calculation after refractive surgery. R Factor has been recently improved [5], so we wonder why the updated formula was not evaluated among the other no-history methods when the accuracy of the ISS method was tested.(3)We have some concerns about the statistical analysis: the authors did not calculate the sample size, which is mandatory in this type of study [6]. Moreover, they did not specify if only eyes with corrected distance visual acuity of 20/40 or better were enrolled, because worse acuity could decrease the accuracy of the crucial postoperative refractive error, as per Hoffer et al.’s protocols [6]. The authors did not check the normality of data, and it is not clear how they compared the percentage of eyes within ±0.25, ±0.50 and ±1.00 D of Prediction Error (PE). These percentages should be compared with Cochran’s Q test instead of Fisher’s test [6]. In addition, we wonder why the authors did not utilize the Friedman test with post hoc test to evaluate the Median Absolute Error (MedAE) of the analyzed formula: it should be preferred when evaluating more than 2 formulas [6].(4)We would also comment on the use of the American Society of Cataract and Refractive Surgery (ASCRS) Online Post-Refractive IOL Calculator to perform calculation with other IOL formulas; with this procedure, only the differences between ASCRS-suggested IOL powers and the implanted IOL powers, corresponding the IOL PEs, could be calculated. A limitation of using IOL-PEs is that it is necessary to convert it to the PE: the PE is traditionally calculated with the assumption that 1.00D of IOL-PE produces 0.70D of PE at the spectacle plane. This ratio of 0.70 is an estimate. Moreover, according to most updated protocols regarding IOL power calculation accuracy studies, converting IOL-PE to PEs using a constant factor over the entire Axial Length (AL) range is an error because this factor changes as a function of ocular parameters [6].(5)In addition, the authors did not declare which lens constant was used to perform the formulas’ accuracy comparisons, and they did not perform a constant optimization trough zeroing out the Mean Error (ME) of PE. Constant optimization is required to eliminate any systematic errors [6]. The only statistically significant differences detected by the study were between the ISS formula compared to Shammas no-history method and Potvin–Hill Pentacam method (dependent by Shammas formula, as reported also by the authors). We need to point out that Shammas’ method showed a statistically significant difference from zero, meaning that a systematic error persists in this formula. If the ME is different from zero, a lens factor either too high or too low for that patient group was used. Zeroing out the ME is the only proper way to eliminate the bias of the lens factor, so that all the formulas are the same [7].(6)Finally, only a few papers in the literature described IOL power calculation methods using a Scheimpflug system and using the AP surface of the corneal radius of curvature; studies by Saiki et al. [8] were certainly pioneers in this area, but the authors forgot to mention the paper by Rosa et al., where a formula to estimate K before refractive surgery (Kpre) based on the postoperative posterior corneal power was proposed [9].

In conclusion, in IOL power calculation accuracy studies, mainly in post-refractive surgery eyes, combining data from different devices should be avoided. To obtain valid results, these studies should be performed according to the most updated protocols in this area [6], among which constant optimization is the most important.

## Data Availability

Not applicable.

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
