# Peer review of "Comment on Iida et al. Development of a New Method for Calculating Intraocular Lens Power after Myopic Laser In Situ Keratomileusis by Combining the Anterior–Posterior Ratio of the Corneal Radius of the Curvature with the Double-K Method. J. Clin. Med. 2022, 11, 522"

_jcm, 2022, doi:10.3390/jcm11071996_

Round 1
Reviewer 1 Report
Dr Rosa's reply to the published article is correct and corresponds to an expert in the biometric calculation. This type of response contributes to generating very interesting and specialised debates on different topics and gives the right to reply to other authors working on the same subject.
Author Response
Response to Reviewer 1 Comments
Point 1: Dr Rosa's reply to the published article is correct and corresponds to an expert in the biometric calculation. This type of response contributes to generating very interesting and specialised debates on different topics and gives the right to reply to other authors working on the same subject.
Response 1: Thank you to your review.

Reviewer 2 Report
Thanks for this Interesting commentary
-some English language improvements to do.
-please precise some abbreviations.
-the order of the points explained must be clarified or reorder to deliver a take home message for the readers, with more logical progression (according the organisation of the original paper or other)
Author Response
Response to Reviewer 2 Comments
Point 1: Thanks for this Interesting commentary
Response 1: Thanks to your review, because it gives us the opportunity to improve our manuscript.
Point 2: some English language improvements to do.
Response 2: Thank you for your suggestion, the paper was fully revised by a native English speaker.
Point 3: please precise some abbreviations.
Response 3: Thank you for your suggestion, we made the changes.
Point 4: the order of the points explained must be clarified or reorder to deliver a take home message for the readers, with more logical progression (according the organisation of the original paper or other).
Response 4: Thanks to your review, we reordered the manuscript according to the organisation of the original paper and we also reordered the references numbers.
Round 2
Reviewer 2 Report
This R1 sounds better for the reader with more logical progression.
English language was improved.
A little "conclusion" for general conclusion to have "the best formula use" depending on variables known available (K/scheimpflug/constant optimisation or not...) => The aim is to have also a take home message from expert point of view, which delivers additional value to the reader. => either a little paragraph, or a little table with affering references
Author Response
Response to Reviewer 2 Comments
Point 1: This R1 sounds better for the reader with more logical progression.
Response 1: Thanks for your suggestions.
Point 2: English language was improved.
Response 1: Thanks.
Point 3: A little "conclusion" for general conclusion to have "the best formula use" depending on variables known available (K/scheimpflug/constant optimisation or not...) => The aim is to have also a take home message from expert point of view, which delivers additional value to the reader. => either a little paragraph, or a little table with affering references
Response 3: Thank you for your suggestion, we included a conclusion paragraph where we synthesize the most important home messages of this paper.